# Preparation of a Novel Solid Phase Microextraction Fiber for Headspace GC-MS Analysis of Hazardous Odorants in Landfill Leachate

**Zonghao Yu [1,2], Ruipeng Yu [1,3], Shengfang Wu [3], Weijie Yu [4] and Qijun Song [1,*]**

[1] School of Chemical and Material Engineering, Jiangnan University, Wuxi 214122, China; 1909401032@stu.suda.edu.cn (Z.Y.); yuruipeng@jiangnan.edu.cn (R.Y.)
[2] College of Chemistry, Chemical Engineering and Material Science of Soochow University, Suzhou 215123, China
[3] State Key Laboratory of Food Science and Technology, Jiangnan University, Wuxi 214122, China; sfwu@jiangnan.edu.cn
[4] Shanghai XTrust Analytical Instruments Co., Ltd., Shanghai 201613, China; yu.weijie@sh-xintuo.com
[*] Correspondence: qsong@jiangnan.edu.cn

**Abstract:** The practice of odorant analysis can often be very challenging because odorants are usually composed of a host of volatile organic compounds (VOCs) at low concentrations. Preconcentration with solid phase microextraction (SPME) is a conventional technique for the enrichment of these volatile compounds before analysis by headspace gas chromatography-mass spectrometry (GC-MS). However, commercially available SPME products usually bear the defects of weak mechanical strength and high cost. In this work, novel SPME fibers were prepared by a one-pot synthesis procedure from divinylbenzene (DVB), porous carbon powder (Carbon) and polydimethylsiloxane (PDMS). Factors that influence the extraction efficiency, such as extraction temperature, extraction time, salting effects, pH, stirring rate, desorption temperature and time, were optimized. VOCs in landfills pose a great threat to human health and the environment. The new SPME fibers were successfully applied in the analysis of VOCs from the leachate of a cyanobacteria landfill. Quantification methods of major odor contributors were established, and a good linearity (r > 0.998) was obtained, with detection limits in the range of 0.30–0.50 ng/L. Compared to commercial SPME fibers, the new material has higher extraction efficacy and higher precision. Hence, it is suitable for the determination of hazardous odorants of various sources.

**Keywords:** solid phase microextraction; one-pot synthesis; gas chromatography-mass spectrometry; hazardous odorants

## 1. Introduction

Volatile organic compounds (VOCs) are the umbrella term for various organic compounds with a boiling point of 50–260 °C at 1 atmospheric pressure [1]. With the acceleration of industrialization and urbanization, the increasing discharge of atmospheric pollutants poses a great threat to human beings [2–5]. Originated from anthropogenic sources, VOCs are mainly generated from fossil fuel combustion, the chemical industry and various wastes. In particular, landfill produces a large quantity of leachate with complex VOCs, including aromatics, sulfides, acids, oxygenates, nitrogen compounds, halogenated hydrocarbons and terpenes [6,7].

Among these VOCs, sulfides, phenolic pollutants and indole homologues are major hazardous contributors to malodor. To elaborate, dimethyl trisulfide has a "swampy odor" [8], and sulfide and volatile organosulfur compounds (VOSCs) are highly toxic, corrosive, and malodorous compounds, which have adverse effects on animal and human health as well as the environment [9]. The phenolic pollutants and their metabolites in living cells cause mutagenicity and carcinogenicity [10]; long-term human exposure to

phenols may cause abnormal breathing, tremor, coma and even death [11,12]. Furthermore, it was reported that 3-MI could form DNA-adduct [13] and induce glomerular sclerosis, hemolysis, improper oviduct functioning and chronic arthritis [14,15]. It thus follows that VOCs contain many hazardous substances.

Owing to their unpleasant odor and potential detrimental effects on human health, VOC issues have aroused intense interest from the academic community, policymakers and the general public. However, VOC monitoring can be very challenging due to the fact that VOCs are often composed of a large number of organic compounds at very low concentrations. For example, the value of indoor gaseous formaldehyde should be less than 0.08 ppm, according to the World Health Organization (WHO) [16]; total VOCs in a chemical industry park can be as low as 28.4 ppb [17]. Therefore, sensitive detection methods should be adopted for the analysis of VOCs.

So far, VOC monitoring can be classified into olfactory identification and instrumental analysis. For the former, poor precision and inaccuracy of classifying odor has constrained its application. Gas chromatography (GC) and gas chromatography-mass spectroscopy (GC-MS) are widely applied in instrumental analysis. Due to the low concentrations of VOCs, pretreatment methods such as activated carbon adsorption, purge and trap methods and solid-phase microextraction (SPME) are often required [18], which substantially improve the capability of qualitative and quantitative analysis. Among these preconcentration techniques, SPME exhibits superiority over other techniques in its convenience and solvent-free methods. It has become one of the frontier extraction technologies and is extensively employed in environmental, food and clinical analysis [19]. Nevertheless, in the practical operation of commercially available SPME fibers, some drawbacks have emerged, including their limited adsorption efficiency, poor reproducibility, relatively low mechanical strength and high cost [20,21].

To overcome the aforementioned limitations, this paper aims to synthesize a simple, highly efficient, strong and economical SPME fiber for VOC analysis. The SPME fiber coating was prepared by the one-pot method, with divinylbenzene, porous carbon powder and polydimethylsiloxane as raw materials. The coating layer on the glass fiber was characterized by scanning electron microscopy (SEM), infrared spectroscopy (IR) and thermogravimetric analysis (TGA). Compared with commercially available products, this SPME fiber can be easily prepared with lower cost. More significantly, with the use of the synthesized SPME fiber, the sensitivity for VOC detection is substantially improved, especially for sulfur-containing compounds.

## 2. Materials and Methods

### 2.1. Chemicals and Standards

The commercial SPME fiber 50/30 μm DVB/CAR/PDMS was obtained from Supelco, USA. Dimethyl sulfide-d6 (99.5%), used as an internal standard, was purchased from Sigma-Aldrich Company, Germany. The standard mixture of n-alkane mixture C4 to C24 was obtained from Sigma-Aldrich (St. Louis, MO, USA). DVB particles (5 μm diameter) and carbon particles (10 ± 3 nm in length) were purchased from Aladdin, Shanghai, China. Superior grade NaCl was used after drying at 500 °C for 4 h. Methanol (chromatographic purity) was purchased from Tedia, China. Milli-Q-Direct 8 pure water meter (Milli-Q, Billerica, MA, USA) and a 20 mL headspace flask (Guangzhou Ingenious Laboratory Technology Co., Guangzhou, China) were also used. All other chemicals were of analytical grade (purity > 98%) from Anpel, Shanghai, China, and were used without further purification.

The preparation of the mixed standard solution was as follows: 100 mg/L of mixed standard solution was prepared as the stock solution and was then diluted to 1.0 mg/L standard solution with water. The preparation of the headspace standard mixed solution was as follows: an appropriate amount of standard solution and internal standard was added to pure water to prepare the standard mixed solution. The actual concentrations of 5.0, 10.0, 40.0, 200 and 1000 ng/L standard mixture solution were prepared with 50.0 ng/L internal standard.

### 2.2. Instruments and Conditions

To study the sub-micron surface features of the SPME coatings, the coatings were examined by a scanning electron microscope (FE-SEM, Hitachi, Tokyo, Japan, S-4800) under an operating voltage of 2.0 kV. FT-IR spectra of the samples were obtained from a Smart iTR diamond ATR in a Nicolet Nexus 470 FT-IR spectrometer (Thermo Electron, Waltham, MA, USA), equipped with a DTGS detector. The spectra were recorded at the absorbance mode from 4000 to 650 cm$^{-1}$ at a resolution of 4 cm$^{-1}$ with 32 scans. For thermo-gravimetric analysis (TGA), an amount of 5.0 mg film sample was weighed into a crucible and was scanned using a thermo-gravimetric analyzer (TGA2, Meteler, Zurich, Switzerland) from 25 to 800 °C at a rate of 10 °C/min in nitrogen atmosphere at a flow rate of 50 mL/min. Heating and stirring was carried out by a magnetic stirrer (C-MAG HS 7 IKAMAG, IKA, Königswinter, Germany). pH measurements were conducted on a Mettler-Toledo pH meter (FiveEasy 20, Suzhou, China) with a combined glass electrode. A gas chromatography time-of-flight mass spectrometer (LECO, St. Joseph, MI, America), PAL RTC automatic sampling system, incubator and heating agitation module (CTC analytics AG, Zwingen, Switzerland) were used. The solid phase microextraction head comprised 120 μm divinylbenzene/carbon/polydimethyl siloxane (DVB/CAR/PDMS). The gas chromatography conditions were as follows: Agilent 7890 B, column DB-5 MS, 30 m × 0.25 mm × 0.25 μm. The temperature programming was as follows: the initial temperature of 40 °C was held for 2 min; then it was increased to 250 °C at 10 °C/min, which was held for 6 min. The following conditions were applied: transmission line temperature: 280 °C; ion source temperature: 210 °C; mass scanning range: 33–400 amu; carrier gas He (99.999%), constant current mode, flow rate: 1.0 mL/min; no split injection was used.

### 2.3. Preparation of the DVB/Carbon/PDMS Coating by One-Pot Synthesis

Hexane was used for the homogeneous dispersion of different types of particles. An appropriate amount of DVB and CAR were weighted and added to hexane to form a mixture with a ratio of 20/40/100 (*w/w/w*). The suspension was initially sonicated for 5 min and then continuously stirred for 60 min. After thoroughly dispersing the particles in the hexane, PDMS was added (20/100, *w/w*, mixture/PDMS). The obtained mixture was then stirred for another 60 min in order to evenly blend the PDMS with the particle suspension. The mixture was purged with nitrogen to evaporate the excessive hexane, which may affect the viscosity of the solution and the thickness of the membrane. Next, a curing reagent (10/100, *w/w*, curing reagent/PDMS) was added into the obtained mixture, which was thoroughly mixed and continuously stirred for 30 min. Finally, the glass fiber was passed through the above mixture steadily to obtain a uniform coating. After coating, the prepared membrane was cured in oven at 120 °C for 4 h.

### 2.4. Solid Phase Microextraction Method

An aliquot of 8.0 mL of the sample solution and 2.0 g of NaCl were placed into the headspace flask and capped tightly with a PTFF/silica gel cover. The flask was immersed in a thermostatic water bath at 35 °C, with stirring at 500 rpm for 10 min till temperature equilibrium. Afterwards, the SPME fiber was inserted into the headspace above the sample. The extraction was conducted for 30 min. Then, the fiber was transferred into a gas chromatography's injector and remained for 5 min at 250 °C in the splitless mode, where the analytes were thermally desorbed and then analyzed using the GC-MS instrument. Each sample was subjected to triplicate injections.

### 2.5. Qualitative and Quantitative Analysis of Volatile Compounds

Qualitative analysis of volatile compounds: the total ion current diagram was automatically deconvoluted by LECO software, ChromaTOF (version 5.40.12.0.60635). After matching with NIST 2017 and Wiley 9, the volatile compounds can be identified only when (1) the positive and negative matching degree is greater than 800 (the maximum value is 1000); (2) the retention index deviation is smaller than 30. Under the same chromatographic

conditions, the retention index (RI) of VOCs calculated by a homologous series of a n-alkane solution (C6–C26) was compared to the standard value of the spectrum library in the limit of tolerance. RI was calculated using the following equation [22]:

$$RI = 100n + 100 \ \frac{t_{R(x)} - t_{R(n)}}{t_{R(n+1)} - t_{R(x)}} \tag{1}$$

where $n$ and $n + 1$ are the numbers of carbon atoms in the eluting n-alkanes before and after the product is produced, $t_{R(x)}$ is the retention time of the compound, $t_{R(n)}$ is the retention time of the heading n-alkane, and $t_{R(n + 1)}$ is the retention time of tailing n-alkane; $t_{R(n + 1)} > t_{R(x)} > t_{R(n)}$.

The peak area normalization method was used to calculate the relative content of the volatile compounds. The concentration of major odor compounds is determined by internal standard method.

## 3. Results and Discussion

### 3.1. Characterization of the Prepared SPME

#### 3.1.1. Structure of Coating

SPME is a technique that separates and preconcentrates the target analytes on polymer coatings from the matrix in one step [23]. The effectiveness of extraction using SPME can be ascribed to its physico-chemical characteristics and structure. It is the ability of the fiber coating that strongly affects the sample preconcentrating effect. In this study, we report a simple, one-pot synthesis of a hybrid coating preparation consisting of PDMS, DVB and carbon particles. The high efficiency and stable extraction ability of the coating is attributed to the uniformity of the coating. It should be emphasized that DVB and carbon particles are highly dispersed in PDMS during the synthesis process.

FTIR was used to determine the functional groups present in the mixed material [24]. In Figure 1, the peaks at 2960 cm$^{-1}$ are attributed to the C-H stretching vibration peak of Si-CH$_3$, and the peaks at 1409 cm$^{-1}$ demonstrate the C-H deformation vibrational peak of Si-CH$_3$ in PDMS. There is no stretching vibration absorption peak of alkene C=C in the region of 1680–1620 cm$^{-1}$, indicating that the polymerization is basically complete. The peaks at 1260 and 842 cm$^{-1}$ are assigned to Si-CH$_3$ deformation, while the wide and strong absorption bands at peaks at 1100–1000 cm$^{-1}$ are related to the Si-O-Si asymmetric stretching vibration. In addition, the peaks at 787 and 740 cm$^{-1}$ are assigned to Si-CH$_3$ oscillation and Si-C stretching vibration, which are characteristic in PDMS. Based on the FTIR spectral analysis, it was confirmed that the SPME was successfully synthesized. Compared with the FTIR diagram of commercial SPME fibers (Figure S1), both diagrams show peaks at nearly the same position, indicating that both fibers bear the same functional groups as shown in IR. This observation can be further confirmed in that the only difference in composition between commercial fibers and synthesized fibers is the difference between CAR and Carbon, both of which show no characteristic peaks in the IR spectrum.

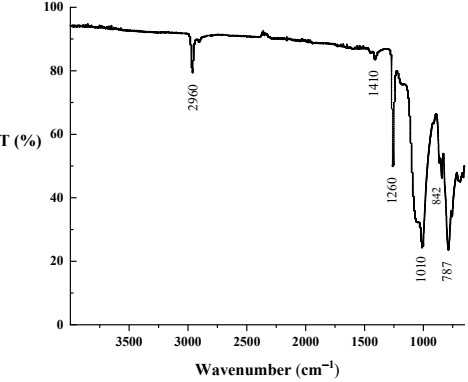

**Figure 1.** FTIR spectra of prepared DVB/Carbon/PDMS SPME coating.

### 3.1.2. Thermal Stability of Coating

TGA was conducted to test the thermal stability of SPME coatings [25]. If the synthesized material has a higher decomposition temperature than the analyte, then the material is suitable for SPME coatings. The TGA curve is shown in Figure 2a. Weigh loss between 60 to 120 °C can be explained by the evaporation of small molecules such as water and methanol, which reside in DVB/Carbon/PDMS during synthesis. The weight loss at 250 °C for 60 min was merely 0.558%. It thus follows that such coating material bears sound thermal stability and is able to meet the thermal desorption standard of SPME analytes. Figure 2b depicts the change in the percentage of the weight of the material, which does not undergo decomposition until 320 °C.

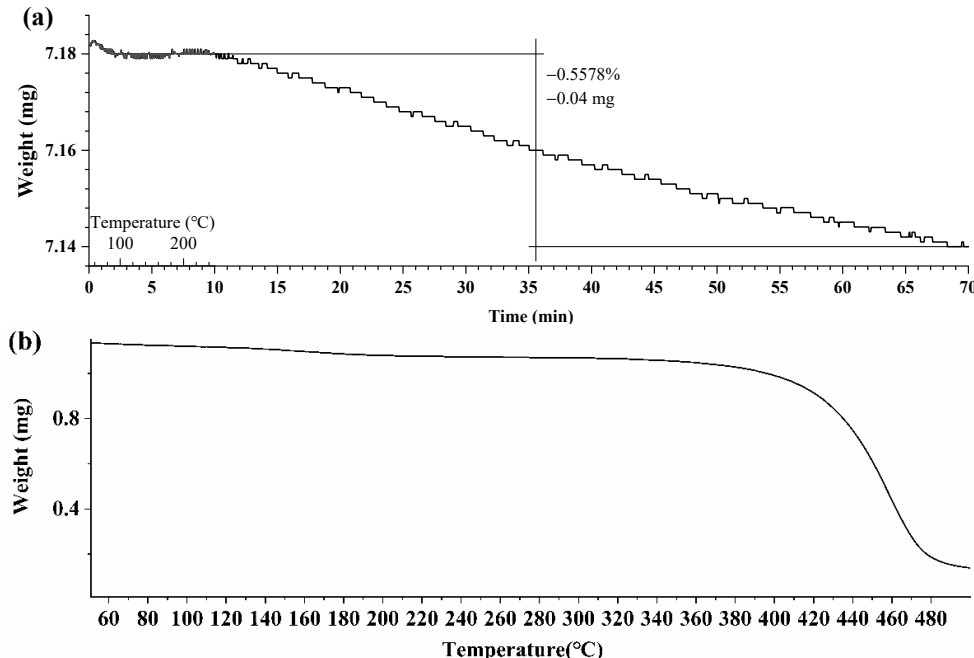

**Figure 2.** TGA of prepared DVB/Carbon/PDMS SPME fiber: (**a**) temperature programming: initial temperature 50 °C, heating rate 20 °C/min, 20~70 min, constant temperature 250 °C; (**b**) temperature range: 50 °C~500 °C.

### 3.1.3. Surface Morphology of Coating

SEM was used to characterize the morphology of the prepared SPME fiber [25]. Figure 3a is a photo of the fiber with a uniform and complete surface. The coating does not fall off under external force extrusion. Figure 3b,d shows images taken from both the surface and the cross section of the fiber (400×). The coating thickness of the prepared SPME was measured at 90 μm, which is higher than the conventional SPMEs (such as PDMS, 75 mm). Moreover, the surface of the synthesized SPME was rougher in comparison to SPME synthesized using the traditional approach, resulting in an extension of surface area and consequently a higher adsorption efficiency. In addition, the cross-section evaluation (Figure 3e) emphasized a porous structure of the inner part of the sorbent coating. Due to the loss of solvent in the rapid thermal fixation process, some micron openings were formed on the surface of PDMS film, rendering the surface of PDMS film rougher (Figure 3c). The DVB and carbon particles in the PDMS were evenly distributed, and no separation layer or aggregation was observed, which ensures the reproducibility of the coating. This result can be attributed to sufficient stirring during membrane preparation. Moreover, the coating prepared by chemical bonding one-pot method can better overcome the defect of the poor mechanical strength of commercial coatings due to their multi-layer coatings.

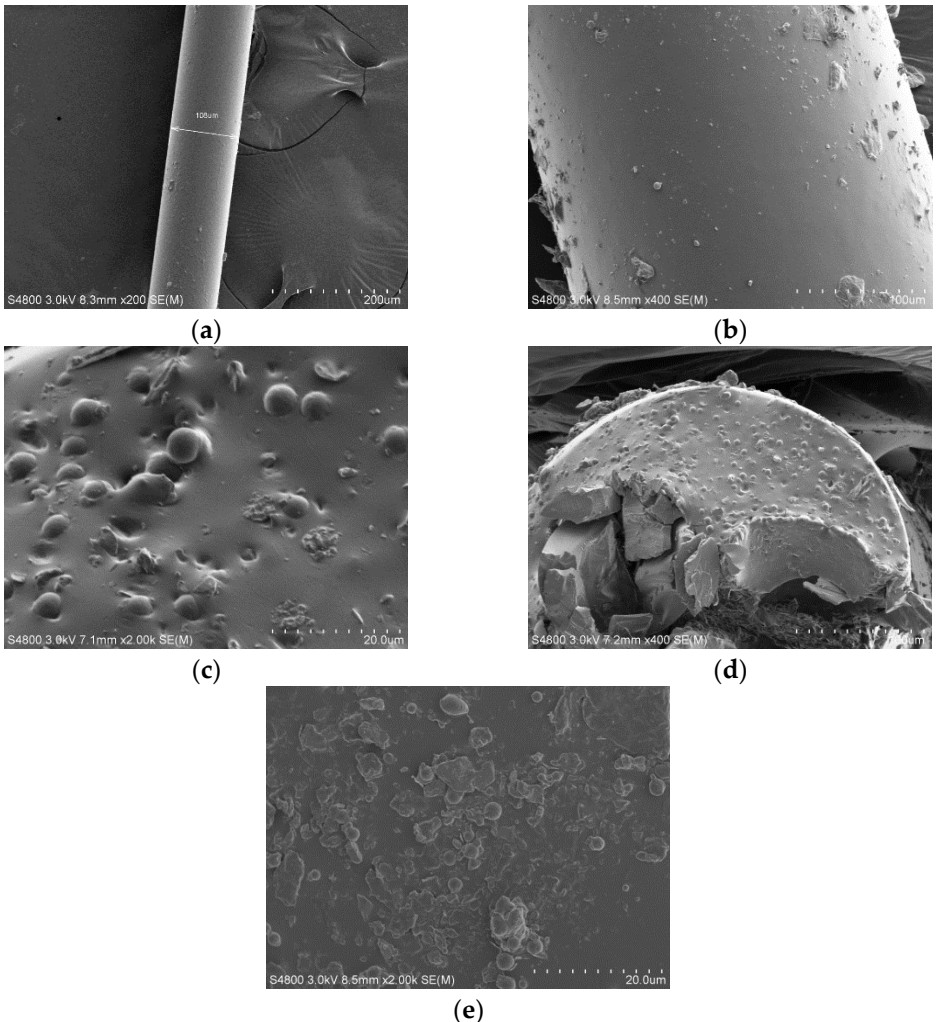

**Figure 3.** SEM micrographs of (**a**) bare fiber, (**b**) prepared fiber at 400×, (**c**) prepared fiber at 2000×, and cross-section of the prepared fiber at (**d**) 400× and (**e**) 2000×.

### 3.1.4. The Effect of SPME Fiber

The determinant property of the fiber coating is the extraction efficacy. Different wrapping materials possess different polarities; based on this, they can selectively absorb volatile compounds according to the "like dissolves like" principle [26]. PDMS is a universal hydrophobic adsorbent. DVB coatings are usually applicable to relatively large molecules and a mono matrix that has several targeted analytes, while carbon centers on molecules of relatively low molecular weight. In this work, the outer layer of commercial DVB/CAR/PDMS is mainly DVB, and the inner layer is mainly CAR/PDMS. Small molecules are first adsorbed inside the DVB coating, while large molecules are adsorbed on its outer surface.

### 3.1.5. Comparison between Synthesized DVB/Carbon/PDMS and Commercial DVB/CAR/PDMS Coatings

The selection of commercial DVB/CAR/PDMS as the comparative object takes many factors into account. Our previous research found that DVB/CAR/PDMS has a competitive edge over DVB/PDMS, PDMS, CAR/PDMS and PA in that DVB/CAR/PDMS fiber has a better extraction effect in the quantification of odorants. Specifically, its extraction rate is ten times higher than that of PA [27].

The prepared DVB/Carbon/PDMS and commercial DVB/CAR/PDMS were compared in their extraction effects on VOCs from leachate. The total ion chromatogram is

shown in Figure S2. Under different experimental conditions, the extraction efficiency was evaluated by the total peak area (TA), sulfide peak area (SA) and the number of volatile organic compounds (TN) obtained from total ion chromatography. The TA and SA of the self-prepared DVB/Carbon/PDMS were 1.69 and 1.80 times higher than that of the commercial DVB/CAR/PDMS, respectively, which means that the extraction efficiency of self-made SPME coatings is higher than that of the commercial one. Furthermore, 91 VOCs were identified by GC-MS, and their name, formula, CAS number, KI value and peak area are listed in Table S1. The peak areas obtained by the self-prepared SPME fiber are larger than by the commercial ones, indicating a higher extraction efficacy of self-made DVB/Carbon/PDMS. Compounds dimethyl trisulfide, dimethyl disulfide, p-cresol, phenol and 3-methylindole are VOCs with high concentrations and are regarded as major hazardous materials in terms of malodor. They are seen as typical VOCs and are further quantified in Section 3.3. Their corresponding MS spectra are provided in Figures S3–S7.

### 3.2. Optimization of HS-SPME

In order to optimize the conditions of VOC analysis by SPME, the effects of extraction temperature, extraction time and the salting out procedure on the efficiency of SPME were investigated. In addition, the efficacy of the SPME-GC system is also dependent on desorption temperature and time. After every three samples, the fiber blank unexposed to the sample was run with optimized parameters to ensure that under the experimental conditions, all analytes were removed before further analysis so as to avoid residual contamination.

#### 3.2.1. The effect of Extraction Temperature

In the extraction process, VOCs need to be evaporated from the sample solution to the gas phase by heating the sample, but with the increase of temperature, the SPME coating/headspace partition coefficient decreases. Suitable extraction temperature enables solid phase microextraction to stably adsorb more VOCs. The effect of temperature on extraction efficiency was investigated under the temperature of 25, 35, 45 and 55 °C and with an extraction time of 30 min. As shown in Figure 4a, with the increase of temperature, more volatile components were detected, and the extraction efficiency was generally enhanced. When the temperature exceeded 35 °C, however, the TN remained basically unchanged, TA increased slightly and SA decreased gradually. It is known that the adsorption on fibers is an exothermic process [26]. When temperature exceeded 35 °C, although the peak area of VOCs were increased to some extent, the total number of volatile compounds remained unchanged, and the sulfur adsorption capacity of the fiber decreased sharply, indicating that the sulfur adsorption was inhibited at an elevated temperature. Therefore, 35 °C was selected as the optimal heating temperature for extracting volatile compounds from landfill leachate, and the balance between volatility and absorption was achieved.

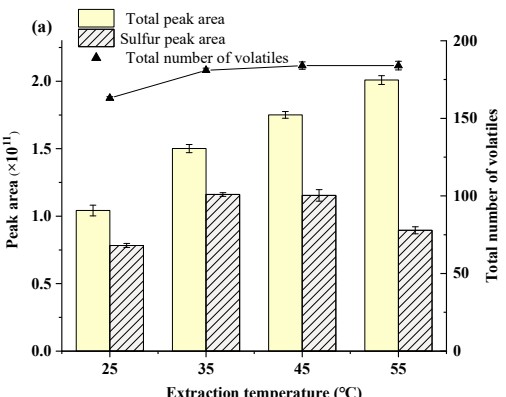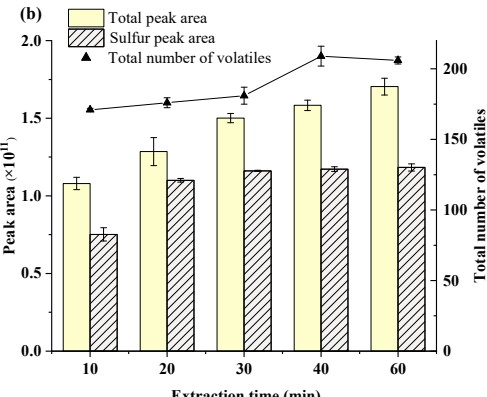

**Figure 4.** Effect of (**a**) extraction temperature and (**b**) extraction time on the extraction efficiency of the VOCs.

### 3.2.2. The Effect of Extraction Time

The extraction time affects the distribution of VOCs between the sample solution and SPME coating and hence directly affects the analytical performance. The extraction time is related to the volatility of compounds and the adsorption dynamics of the SPME fiber [28]. A longer extraction time is conducive to the volatiles' occupying more space on the fiber, but prolonging the occupation time of all positions will not improve the preconcentration efficiency. Conversely, it could even trigger desorption [29]. However, an extraction equilibrium is not necessarily reached in order in an effective extraction, and a balance between the sensitivity and extraction time must be considered. Figure 4b shows that TA, SA and TN increase steadily with the extension of extraction time from 10 to 30 min. There was no significant difference in SA values obtained from 30, 40 and 60 min. Therefore, 30 min was considered to be the optimal extraction time.

### 3.2.3. The Effect of Salt Contents on the Extraction

The salinity of the sample solution also affects the efficiency of SPME. It was reported that the salt effect improves the ionic strength of the solution, reduces the solubility of the target compound in the solution and promotes the volatilization of malodorous substances [30]. In the present work, the salt effect was studied by adding a series of 0–1.8 g NaCl to 6.0 mL of sample solution. As can be seen from Figure 5a, the addition of NaCl significantly increased TN and TA in comparison to that of the control group. These results suggest that a certain amount of salt is helpful to drive volatiles from the sample matrix to the fiber coating. The increase in the total number of volatile compounds and their peak area can be attributed to the fact that the addition of NaCl can enhance the ionic strength of water samples [31], thereby reducing the solubility of organic compounds by the so-called salting out effect. With the increase of salt concentration, however, the extraction efficiency for SA increases at first and then decreases gradually. It was found when the concentration of NaCl reached 20%, the peak area of the SA tended to be stable. Thus, this concentration was selected for the subsequent experiment.

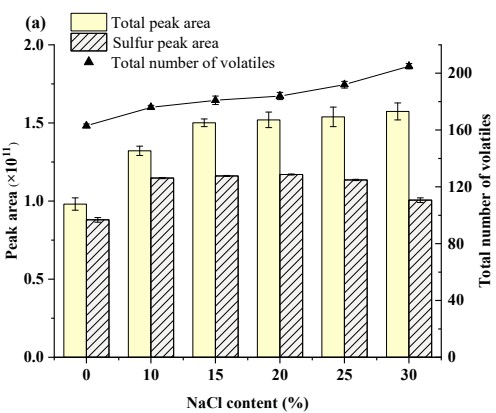 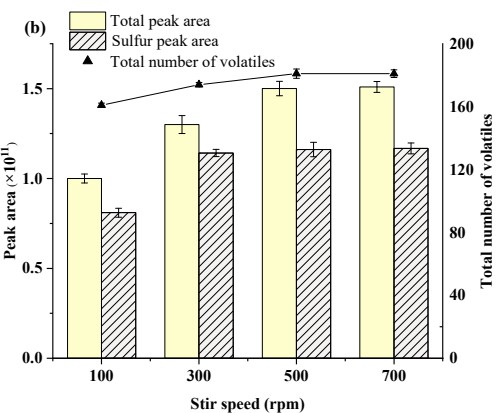

**Figure 5.** Effect of (**a**) NaCl and (**b**) stir speed on the extraction efficiency of the VOCs.

### 3.2.4. The Effect of pH Value on Extraction

The effect of pH on extraction efficiency was studied for a pH of the solution of 3, 5, 7, 9 and 11. The results illustrated that variation in pH value has a huge impact on TA, which is probably related to neutral species in alkyl sulfides. Since the pH value of wastewater normally falls between seven and eight, measurement of actual samples does not require adjusting their pH value.

### 3.2.5. Stirring Speed

Stirring the sample can accelerate the diffusion of analytes from the aqueous solution to the SPME coating solution, so as to improve the extraction efficiency and shorten the

extraction time. In this paper, the effect of stirring speed (100–700 rpm) on extraction efficiency was studied. As shown in Figure 5b, when the stirring speed was increased to 500 rpm, the extraction efficiency of VOCs increased. When the stirring speed was further increased, the extraction efficiency remained unchanged. Furthermore, when the stirring speed was greater than 500 rpm, the stirring rod became unstable. Therefore, 500 rpm was selected as the optimal stirring speed.

### 3.2.6. The Effect of Desorption Temperature

The complete desorption of VOC pre-adsorption on the SPME coating is the key to GC-MS analysis. The desorption temperature and time were optimized, and the results are shown in Figure 6. Increasing the desorption temperature led to shorter desorption time of the VOCs from the SPME. Considering the high volatility of the compounds of the leachate of landfill, a fiber desorption temperature between 220 and 280 °C was tested, with a constant desorption time of 5 min. Since most components are completely desorbed at 250 °C, it was selected as the optimum desorption temperature (higher temperatures are not recommended so as to avoid siloxane compounds shedding from SPME).

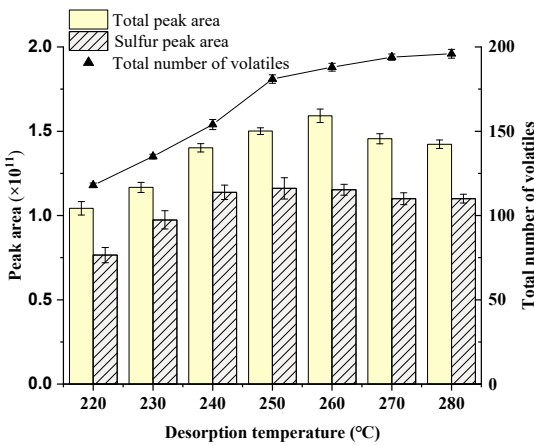

**Figure 6.** Effect of desorption temperature on the extraction efficiency of the VOCs.

### 3.2.7. The Effect of Desorption Time

Optimizing the desorption time can prevent the insufficient relaxation caused by an in-adequate desorption time, which affects the subsequent extraction. Alenyorege et al. [32] investigated the desorption time in six time periods (2–7 min). Some components were not completely desorbed within 2 min. With the extension of the desorption time, the total peak area, main peak area and total peak number increased and reached the platform at 5 min, indicating that the best resolution time is 5 min. In addition, the TN and TA of volatile compounds did not show obvious distinction from 4 to 7 min, and the analyte in the fiber showed sufficient desorption within 5 min. It could be inferred that highly volatile compounds can be decomposed within 2 min because they are scarcely affected by the desorption time. The low volatile composition can be decomposed gradually with the extension of analytical time, and the sulfur-containing compounds can be completely decomposed within 5 min.

### 3.2.8. The Effect of Sample Volume

HS-SPME technology is based on the dynamic balance of analytes among samples, headspace and fiber coatings [33]. Therefore, sample volume will affect the ionic strength and vapor pressure of volatile analytes, thereby affecting the extraction efficiency. The experimental results showed that some compounds are low in content, so it should be avoided when the sample amount is too small to be adsorbed by the fiber. However, high sampling content is also unfavorable, owing to the fact that fiber easily becomes saturated and absorbed, and the excessive volatile compound will not be continuously collected.

According to the experimental results, 8 mL was selected for subsequent investigation. Furthermore, in authentic water sample analysis, samples with a high content of VOCs are diluted into an appropriate quantitative region.

### 3.3. Quantitative Analysis of Volatile Compounds

A series of headspace standard mixed solutions at concentration levels of 5, 10, 40, 200 and 1000 ng/L were prepared. The standard curve was drawn under the above-mentioned optimized conditions. The calculated calibration curves gave a high level of linearity for the target compounds in the range of 5–1000 ng/L, with averaged correlation coefficients ($R^2$) ranging between 0.9951 and 0.9986. The detection limit (LOD) was calculated using three times the signal-to-noise ratio. In addition, the limits of quantitation (LOQs) were calculated using ten times the signal-to-noise ratio.

The method linear equation, correlation coefficient and detection limit are shown in Table 1. It is evident that the preparation of the fiber by the proposed method has good reproducibility. These results showed that the optimized HS-SPME method was capable of achieving acceptable precision for the determination of volatile compounds in leachate.

**Table 1.** Linear equation, correlation coefficient, limit of detection and RSD of the method.

| Analyte | Quantification Ions | Linear Range (ng/L) | Linear Equation | $R^2$ | LOD [a] $\rho$/(ng/L) | LOQ [b] $\rho$/(ng/L) | RSD(%) $n = 6$ |
|---|---|---|---|---|---|---|---|
| dimethyl trisulfide | 126 | 10–1000 | y = 0.2533x − 0.2253 | 0.9935 | 3.0 | 10.0 | 2.8 |
| dimethyl disulfide | 94 | 10–1000 | y = 0.0956x − 0.1167 | 0.9956 | 5.0 | 20.0 | 3.3 |
| p-Cresol | 107 | 5–1000 | y = 0.9674x + 0.2662 | 0.9948 | 1.0 | 3.0 | 4.5 |
| Phenol | 94 | 5–1000 | y = 0.1534x + 0.1387 | 0.9987 | 1.0 | 3.0 | 8.7 |
| 3-methylindole | 131 | 5–1000 | y = 0.0712x − 0.0034 | 0.9992 | 1.0 | 3.0 | 3.6 |

[a] Limit of detection for S/N = 3; [b] Limit of quantification for S/N = 10.

### 3.4. Application of SPME Fiber for Real Sample Analysis

The leachate from different processes of waste stations was analyzed with prepared SPME. Table 2 lists the quantitative results of water samples. The results show that the concentration of odor compounds in the initial collection tank was the highest, which was then passed through the regulation tank. After the reaction tank, the concentration of the final outlet tank decreased greatly. In order to evaluate the accuracy of this method, 200 ng/mL volatile organic compounds were added to the sample from the outlet pool. In the test of recovery, the recovery rate of the sample was 82.7~105.4%. The results of relative recovery and relative standard deviation obtained by repeated analysis showed that the matrix had no significant effect on the efficiency of solid phase microextraction.

**Table 2.** Detection results and recoveries of volatile metabolites in water samples ($n = 3$).

| Compounds | Collecting Pool (ng/L) | Regulating Tank (ng/L) | Reaction Tank (ng/L) | Outlet Pool (ng/L) |
|---|---|---|---|---|
| Dimethyl trisulfide | $1.54 \times 10^6$ | $4.28 \times 10^4$ | $5.86 \times 10^2$ | 17.0 |
| Dimethyl disulfide | $3.23 \times 10^6$ | $4.53 \times 10^4$ | $6.34 \times 10^2$ | 22.3 |
| p-Cresol | $5.23 \times 10^5$ | $6.43 \times 10^3$ | $5.26 \times 10^2$ | 7.65 |
| Phenol | $8.23 \times 10^5$ | $9.36 \times 10^3$ | $7.96 \times 10^2$ | 9.03 |
| 3-Methylindole | $1.03 \times 10^4$ | $6.38 \times 10^2$ | 84.5 | 2.51 |

## 4. Conclusions

A simple and economical SPME fiber was synthesized by the one-pot method, with divinylbenzene, porous carbon powder and polydimethylsiloxane as raw materials. The

results showed that the synthesized coating has good thermal stability and high extraction efficiency. The SPME fiber was applied to the analysis of volatile organic compounds in landfill leachate, and a total of 91 volatile compounds were identified, including sulfide, aromatics, alcohols, nitrogen compounds, esters, alkanes, ketones, terpenes, acids, aldehydes and so forth. Compared with the commercial fiber (DVB/CAR/PDMS), the self-made fiber (DVB/Carbon/PDMS) had higher quantity and content of extract, and it was simpler in preparation and cost-efficient. For the analysis of five typical hazardous substances, in the concentration range of 10.0–1000 ng/L, a good linearity (r > 0.998) was obtained, with their detection limits in the range of 0.30–0.50 ng/L, and recoveries at the concentration level of 50 ng/L were 76.3–93.0%. The new method exhibited good precision, with RSD values of less than 12.7%. Therefore, the prepared solid phase microextraction fiber can be used for rapid detection and analysis of hazardous odorants in the environment.

**Supplementary Materials:** The following supporting information can be downloaded at: https://www.mdpi.com/article/10.3390/pr10061045/s1, Figure S1: A comparison of FT-IR diagram of: (a) commercial PDMS/CAR/DVB, (b) self-made PDMS/Carbon/DVB; Figure S2: Amplified total ion chromatograms of VOCs from landfill leachate obtained from: (a) prepared PDMS/Carbon/DVB fiber; (b) the commercial DVB/CAR/PDMS fiber. (Upper-right: overall view); Figure S3: Mass spectrum of dimethyl disulfide: (a) standard spectrum from the library; (b) separated by GC-MS from aquatic samples; Figure S4: Mass spectrum of dimethyl trisulfide: (a) standard spectrum from the library; (b) separated by GC-MS from aquatic samples; Figure S5: Mass spectrum of p-cresol: (a) standard spectrum from the library; (b) separated by GC-MS from aquatic samples; Figure S6: Mass spectrum of phenol: (a) standard spectrum from the library; (b) separated by GC-MS from aquatic samples; Figure S7: Mass spectrum of 3-methylindole: (a) standard spectrum from the library; (b) separated by GC-MS from aquatic samples; Table S1: VOCs results of landfill leachate obtained from the prepared and commercial SPME.

**Author Contributions:** Conceptualization, Z.Y. and R.Y.; methodology, Z.Y.; formal analysis, Z.Y.; investigation, W.Y.; pretreatment, Z.Y. and W.Y.; data curation, S.W.; writing, Z.Y.; supervision, R.Y. and Q.S.; project administration, R.Y.; funding acquisition, Q.S. All authors have read and agreed to the published version of the manuscript.

**Funding:** This research was funded by the National Natural Science Foundation of China (No. 51973083) and National First-Class Discipline Program of Food Science and Technology (JUF-STR20180301).

**Institutional Review Board Statement:** Not applicable.

**Informed Consent Statement:** Not applicable.

**Data Availability Statement:** Not applicable.

**Acknowledgments:** The authors acknowledge Junkang Liu for his contribution to SEM study and acknowledge Xiaohong Gu for her support in IR.

**Conflicts of Interest:** The authors declare no conflict of interest.

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
