# Peer review of "Preparation of a Novel Solid Phase Microextraction Fiber for Headspace GC-MS Analysis of Hazardous Odorants in Landfill Leachate"

_processes, doi:10.3390/pr10061045_

Round 1

Reviewer 1 Report

This manuscript discusses issues related to volatile organosulfur compounds (VOC). Methods for recording these volatile substances, their effect on humans, methods of concentration, etc. are discussed. The work was done to a decent standard. The language of presentation is simple and understandable. The materials and methods used are described in detail and exhaustively. The topics covered are quite specific and most likely will be of interest primarily to a narrow circle of specialists. However, the average reader can find interesting facts in the submitted manuscript. For example, allowable concentrations of volatile substances. I recommend improving the quality of the figures, this will make them more informative. The following is a list of minor suggestions and recommendations:   Line 15. It is necessary to decipher all acronyms at the first mention. Line 24. "fiber" should be replaced with "fibers" Figure 1. Wavenumber I recommend rounding the values. Figure 2. Add decoding for "a" and "b" to the caption Figure 3. Improve the quality of the scale bar in photographs. Figures 9, 10. A unit must be specified for the peak area.

Author Response

Thank you for the instructive advice given in the peer review. Here is my response (The modifed article can be viewed in the attached Word document):

Point 1. Line 15. It is necessary to decipher all acronyms at the first mention.

Full names are given in the modified article. 

Point 2. Line 24. "fiber" should be replaced with "fibers".

Plural forms are taken in the modified article.

Point 3. Figure 1. Wavenumber I recommend rounding the values.

Wavenumbers have been rounded in the modified article.

Point 4. Figure 2. Add decoding for "a" and "b" to the caption.

Fig. 2a and Fig. 2b have been explained in the modified article.

Point 5. Improve the quality of the scale bar in photographs.

SEM figures have been replaced with clearer forms in the modified article.

Point 6. Figures 9, 10. A unit must be specified for the peak area.

(1) The author has checked the GC-MS operation interface, and the column "Peak area" does follow with a unit.

(2) The author referred to "Instrumental analysis" textbook and found that the verticle axis of total ion chromatograms stands for intensity, which does not have a unit. It can be inferred that peak area, the integration of intensity with time, should also give no units.

Reviewer 2 Report

The manuscript entitled Preparation of a novel solid phase microextraction fiber for headspace GC-MS analysis of hazardous odorants in landfill leachate is a very interesting topic related to the extraction and characterization of volatile compounds. The information in the manuscript is well written and presented. There are some minor details that authors must attend to prior to publication. Below are the comments.

-Improve the quality of the figures.

-Did the authors carry out an FT-IR analysis for the commercial DVB/CAR/PDMS fiber as a control? It must be interesting to compare the results.

-The authors presented in figure 4 the total ion chromatograms of VOCs for prepared and commercial fibers. However, the image by itself indicates nothing. Also, table 1 presents the identified VOCs, but it does not indicate if they were extracted using the prepared or the commercial fibers. I suggest including figure 4 as supplementary material and indicating in table 1 which compounds were extracted by using the commercial or the prepared SPME.

Author Response

Thank you for your instructive advice, my resoponse is as follows (modifed article can be viewed in the attached Word document):

Point 1. Improve the quality of the figures.

SEM figures have been replaced with clearer figures in the polished article.

Point 2. Did the authors carry out an FT-IR analysis for the commercial DVB/CAR/PDMS fiber as a control? It must be interesting to compare the results.

FT-IR spectrums of commercial fibers and the synthesized ones have been put in SI, and comments have been added in the main text. Both display peaks at relatively the same position, which is quite reasonable due to the fact that their only difference in compostion, CAR and Carbon, do not show characteristic peaks at detected wavenumbers.

Point 3. The authors presented in figure 4 the total ion chromatograms of VOCs for prepared and commercial fibers. However, the image by itself indicates nothing. Also, table 1 presents the identified VOCs, but it does not indicate if they were extracted using the prepared or the commercial fibers. I suggest including figure 4 as supplementary material and indicating in table 1 which compounds were extracted by using the commercial or the prepared SPME.

Your advice is pertinent. We have put the total ion chromatogram in SI. For table 1, A and B are peak area of extracted componets by synthesized fibers and commerical fibers, respectively. They have also been clarified in the polished article.

Reviewer 3 Report

The paper is well-written. Few minor changes are suggested before accepting for a publication.

1. Please indicate the method detection limits in the abstract

2. Table 1 may be moved to the supporting material

3. SEM figure scale bars are not clear to the reader

4. Referencing is required for the SPME fiber characterization section.

5. Please label the key peaks in IR spectra

6. The figures appear to be in low-resolution. At least 300 DPI is required for figures. Also, figure axes fonts sizes needs to increased and bold fonts are recommended.

7. Figure 4 - TIC is cropped from the top and please fix this. Also, I have not seen any mass spectra corresponding to each retention time. At least it is important to include those for dimethyl tri/di sulfide, p-Cresol, Phenol and 3-methyllindole.

8. Figures 5-10 - Some of these can be combined together

9. Line 400 - change "real" to "authentic water"

10. Line 406 - change "ng/mL-1" to "ng/mL"

11.  Authors claim that the fiber preparation is economical and it is suggested to give a cost analysis of preparation and compare it with a price of a commercial SPME fiber pack.